# Release of Bioactive Peptides from *Erythrina edulis* (*Chachafruto*) Proteins under Simulated Gastrointestinal Digestion

**DOI:** 10.3390/nu14245256

**Published:** 2022-12-09

**Authors:** Jessica L Correa, José Edgar Zapata, Blanca Hernández-Ledesma

**Affiliations:** 1Nutrition and Food Technology Group, Universidad de Antioquia, Medellin 050010, Colombia; 2Development and Innovation in Alternative Proteins Group, Institute of Food Science Research (CIAL, CSIC-UAM, CEI UAM+CSIC), Nicolás Cabrera, 28049 Madrid, Spain

**Keywords:** *Erythrina edulis*, simulated gastrointestinal digestion, multifunctional peptides, immunomodulatory activity

## Abstract

The estimated and concerning rise in world population over the next few years and the consequent increase in food demand will lead to a deterioration in global food security. To avoid or reduce this world crisis, informed and empowered consumers are turning to sustainable and nutrient-rich foods that substitute animal products, also reducing their associated environmental impact. Moreover, due to the demonstrated influence of diet on the risk of high incidence and mortality of noncommunicable diseases, the current established food pattern is focused on the consumption of foods that have functionality for health. Among these new foods, traditional and underutilized plants are gaining interest as alternative protein sources providing nutritional and biological properties. In this work, the potential of *Erythrina edulis* (*chachafruto*) proteins as a source of multifunctional peptides after transit through the gastrointestinal tract has been demonstrated, with antioxidant and immunostimulating effects in both biochemical assays and cell culture. While low molecular weight peptides released during the digestive process were found to be responsible for protection against oxidative stress mediated by their radical scavenging activity, high molecular weight peptides exerted immunostimulating effects by upregulation of immunoresponse-associated biomarkers. The findings of this study support the promising role of chachafruto proteins as a new antioxidant and immunostimulatory ingredient for functional foods and nutraceuticals.

## 1. Introduction

In the last few years, the estimated rising of the world population, the consequent deterioration in global food security, and the concern about the environmental impact of food systems have made international organizations, governments, industries, and societies recommend a dietary shift to reduce the intake of calories from animal sources and increase the consumption of sustainable, nutrient-rich, and calorically efficient products [1]. Moreover, due to the worldwide rise in incidence and prevalence of noncommunicable diseases (NCDs) and the demonstrated impact of diet on the risk of these disorders, the current established food pattern is focused on the consumption of foods that have functionality for health. Because of the interest gained in plant proteins as part of a healthy and sustainable diet, research on traditional and underutilized plants as a source of highly nutritional and bioactive proteins has intensified in the last years [2,3].

*Erythrina edulis* (*chachafruto*) is a legume endemic to some South American countries with a wide range of nutritional and medicinal uses [4]. Its protein content (18–24%) and quality is superior to that reported for other legumes and similar to that of egg proteins [5]. However, although the protein content is high, the data available on the potential of these proteins as a source of bioactive peptides are still scarce. Preliminary studies performed in our laboratory have described the in vitro antioxidant, antidiabetic, and antihypertensive activity of chachafruto protein-derived hydrolysates by microbial or gastric enzymes [6,7]. However, to the best of our knowledge, there are no data on the potential of gastrointestinal digestion simulating physiological conditions to hydrolyze *E. edulis* proteins and release multifunctional peptides.

Simulated gastrointestinal digestion has become one of the most widely studied approaches to generate bioactive peptides from food proteins because it is fast, inexpensive, safe, and does not include ethical restrictions [8,9]. In vitro digestion tests allow releasing peptides encrypted within the protein matrix, simulating the process naturally occurring into the human body. Moreover, it allows understanding of how dietary proteins are degraded under physiological conditions [10]. A static gastrointestinal digestion was successfully conducted to degrade legume proteins such as yellow peas pinto beans and release peptides with antioxidant and/or ACE inhibitory activity [11,12].

Although a majority of the studies have focused on antihypertensive, antimicrobial, antidiabetic or antiproliferative peptides [13,14], the recent research on immunomodulatory peptides with positive impact on the host defense has gained interest [15,16]. However, peptides with immunomodulatory effects represent the most diverse and complex group, being needed for screening different targets (oxidative status, and inflammatory and immunoresponse-associated biomarkers) to confirm their activity. The variety of conditions resulting from compromised function of the immune system and the wide range of bioactivities that immunomodulatory peptides can exert make them promising ingredients for novel functional foods and/or nutraceuticals [17].

Thus, this work aimed at investigating the potential of gastrointestinal digestion simulating physiological conditions to release multifunctional peptides, focusing on their radical scavenging capacity and their protective effects on oxidative stress and immunoresponse-associated biomarkers.

## 2. Materials and Methods

### 2.1. Materials

*E. edulis* (chachafruto) seeds were collected in Rioblanco–Tolima (Colombia). Alcalase 2.4 U/g protein from *Bacillus licheniformis* was obtained from Novozymes (Bagsvaerd, Denmark). Bovine seroalbumin (BSA), 2,2′-azino-bis(3-ethylbenzothiazoline-6-sulfonic acid) (ABTS), 6-hydroxy-2,5,7,8-tetramethylchroman-2-carboxylic acid (trolox), disodium fluorescein (FL), 2,2′-azobis (2-amidinopropane dihydrochloride) (AAPH), fetal bovine serum (FBS), Dulbecco’s Modified Eagle’s Medium/High Modified (DMEM), L-glutamine solution, sodium pyruvate, non-essential amino acids (NEAA), gentamicin, 3-[4,5-dimethylthiazol-2-yl]-2,3-diphenyl tetrazolium bromide (MTT) were acquired from Sigma-Aldrich (St. Louis, MO, USA). All other reagents were of analytical grade.

### 2.2. Seed Flour Analysis

Seeds were cleaned, peeled, cut into tiny pieces, dried at 40 °C, ground in a domestic mill, and filtered using mesh number 60. The proximate composition of the resulting homogenous flour sample was analyzed following the AOAC methods [18] for moisture (934.01), fat (930.09), ash (930.05), crude fiber (934.10), and protein (978.04). The conversion factor to calculate the protein value was 6.25.

### 2.3. Protein Concentrate from Chachafruto Seeds (CPC)

Chachafruto flour was suspended in water (10%, *w/v*), its pH was adjusted to 10.0 by adding 1 N NaOH, and the suspension was magnetically stirred for 2 h at 60 °C, and centrifugated at 4500× *g* for 20 min at 25 °C. The pH of the supernatants was adjusted to 4.5 with 1 N HCl and the suspensions were left at 4 °C overnight. The precipitated proteins were collected by centrifugation at 10,000× *g* for 20 min at 4 °C, resuspended in water, lyophilized, and stored at −20 °C until further analyses. The amino acid content of the chachafruto protein concentrate (CPC) was analyzed by cation exchange chromatography using a Biochrom 30 series Amino Acid Analyser (Biochrom, Cambridge, UK) after automatic post-column derivatization of samples with ninhydrin and measurement of the absorbance at 570 nm. The previous hydrolysis of the samples was performed with 6 M HCl for 21 h at 110 °C. The results were expressed as mean of two replicates (g of amino acid/100 g protein).

The protein profiles of CPC, their digests, and fractions were analyzed by SDS-PAGE using 12% polyacrylamide gels (Bis-Tris CriterionTM XT Precast Gel, Bio-Rad, Hercules, CA, USA). After mixing the samples with sample buffer [60 mM Tris-HCl pH 6.8, 25% glycerol (*v/v*), 2% sodium dodecyl sulfate (SDS) (p/v), 14.4 mM 2-mercaptoethanol, and 0.1% 2-bromophenol (p/v)], they were heated for 5 min at 100 °C and cooled to room temperature before loading onto gels (50 µg of protein/sample). The analysis was performed in a Criterion automated system (Bio-Rad) using the XT MES Running Buffer 20X (Bio-Rad). The electrophoretic migration was carried out with a voltage of 150 V for 45 min. The gels were stained with Coomassie Blue for 60 min and washed with a 10% acetic acid −10% methanol solution for 12 h. Precision Plus Protein Standard Unstained (Bio-Rad) was used as molecular weight protein standard.

### 2.4. Simulated Gastrointestinal Digestion

CPC was digested according to the in vitro harmonized protocol [19] with some modifications. Briefly, 0.5 g of CPC was dissolved in 5 mL of salivary fluid (pH 7.0, 37 °C) for 5 min. Then, 4 mL of simulated gastric juice (pH 3.0, 37 °C) containing pepsin from porcine gastric mucosa (2000 U/mL of digest, EC 3.4.23.1, Sigma-Aldrich) was added and incubated at 37 °C for 120 min. After adjusting the pH of gastric digests (GDCPC) to 7.0 with 1 M NaOH, and intestinal phase started after addition of simulated intestinal fluid composed of porcine pancreas pancreatin (100 U trypsin activity/mL of final mixture, Sigma-Aldrich) and porcine bile extract (10 mM in the final mixture, Sigma-Aldrich). Digestions (in duplicate) were performed at 37 °C in an orbital shaker at 150 rpm. After 120 min incubation, samples were heated at 80 °C for 15 min to inactive enzymes and obtain the gastrointestinal digest (GIDCPC). Samples were freeze-dried and kept at −20 °C until their analysis. A digestion blank (DB) consisting of the enzyme mixture without CPC was prepared.

CPC, GDCPC, and GIDCPC were subjected to ultrafiltration through hydrophilic 10,000 and 3000 Da cutoff membranes (Merck KGaA, Darmstadt, Germany). Fractions of <3 kDa, 3–10 kDa and >10 kDa were lyophilized and kept at −20 °C until use. The bicinchoninic acid method (BCA) (Pierce, Rockford, IL, USA) was used to measure the peptide content of digests and fractions, using BSA as standard protein.

Analysis of peptide mass distribution in GDCPC and GIDCPC was performed by MALDI-TOF mass spectrometry (MS), following the protocol previously described [20]. Samples were spotted on an Anchorchip target (Bruker Daltonik GmbH, Bremen, Germany) with α-cyano-4-hydroxycinnamic acid (α-CHCA) matrix in acetonitrile/water (30:70) containing 0.1% trifluoroacetic acid (TFA) and analyzed on a Bruker Autoflex Speed^®^ (Bruker Daltonik). Mass spectra were obtained in positive reflectron mode by collecting 1000 laser pulses on average. Calibration was made with the Peptide Calibration Standard I and II (Bruker Daltonik).

### 2.5. Antioxidant Activity by Biochemical Assays

The ABTS•+ scavenging activity was determined according to the enhanced protocol described by Re et al. [21]. 180 μL of diluted ABTS•+ solution and 20 μL of either PBS (blank), trolox (25–200 µM) (standard) or sample (at different concentrations) was mixed and incubated for 5 min at room temperature. After that time, the absorbance was measured at 734 nm in a Synergy HTX microplate reader (BioTek Instruments, Inc., Winooski, VT, USA). The trolox equivalent antioxidant capacity (TEAC) was calculated dividing the gradient of the plot of the percentage inhibition of absorbance versus protein concentration of sample by the gradient of the plot for trolox. The results were expressed as µmol trolox equivalent (TE)/mg of protein.

The oxygen radical absorbance capacity (ORAC) was determined according to the protocol described by Hernández-Ledesma et al. [22] with some modifications. A volume of 20 μL of either PBS (blank), trolox (0.2–1.6 nmol) or sample (at different concentrations) was mixed with 120 μL of FL (117 nM) and incubated at 37 °C for 10 min. Then, 60 μL of AAPH (14 mM) was added and the mixture was incubated for 80 min, reading the fluorescence at 485 and 520 nm of excitation and emission wavelengths, respectively, in a FLUOstar OPTIMA plate reader (BMG Labtech, Offenburg, Germany). The equipment was controlled by the FLUOstar Control ver. 1.32 R2 software. Final ORAC-FL value was expressed as µmol TE/mg of protein (mean of three replicates).

### 2.6. Protective Effects in Macrophage RAW264.7

#### 2.6.1. Cell Culture

The mouse macrophage cell line RAW264.7 (American Type Culture Collection, ATCC, Rockville, MD, USA) was grown in DMEM medium supplemented with 10% of FBS (*v/v*) and 1% penicillin/streptomycin (*v/v*). Cells were seeded in 75 cm^2^ culture flasks and incubated at 37 °C in a humidified incubator containing 5% CO_2_ and 95% air. Culture medium was changed every 2 days and subcultures were made by scraping.

#### 2.6.2. Effects on Cell Viability

Cell viability was determined using the MTT assay. RAW264.7 cells were seeded onto 96-well plates at a density of 6 × 10^4^ cells/well in complete medium with 10% FBS and incubated for 24 h at 37 °C. After removing the culture medium, samples were added. In the case of lipopolysaccharide (LPS)-stimulated cells, 20 μL of LPS (100 ng/mL, final concentration) was also added. Once incubated for 24 h at 37 °C, the supernatant was removed, and cells were washed with PBS. Then, an MTT solution (5 mg/mL in PBS) was added and the plate was incubated for 120 min at 37 °C. Once the supernatant was aspirated, insoluble formazan crystals were dissolved in dimethyl sulfoxide (DMSO), and the absorbance was measured at 570 nm in the Multiskan FC plate reader (ThermoTM Scientific, Wilmington, DE, USA). The results were expressed as percentage of the control, considered as 100%.

#### 2.6.3. Effects on ROS Generation

Intracellular ROS levels were quantified according to the method described by LeBel et al. [23] using dichlorofluorescein (DCFH-DA) as probe. RAW264.7 macrophages were seeded onto 48-well plates (4.75 × 10^4^ cells/well) in complete medium with 10% FBS and incubated for 24 h at 37 °C. Once the medium was aspirated, samples were added and cells were incubated for 24 h at 37 °C. In stimulated cells, 20 μL of LPS (100 ng/mL) was also added. After aspirating the supernatant, 100 µL of a solution containing 5 mM DCFH-DA dissolved in Hank’s balanced salt solution (HBSS, Sigma-Aldrich) was added to the wells, incubating the plate at 37 °C for 60 min. The fluorescence was measured in a FLUOstar OPTIMA plate reader (BMG Labtech) at λ_excitation_ of 485 and λ_emission_ of 530 nm. The results were expressed as ROS levels (% compared with the control, considered as 100%).

#### 2.6.4. Effects on NO Levels

RAW264.7 macrophages were seeded onto 96-well plates (1× 10^5^ cells/well) in complete medium with 10% FBS and incubated for 24 h at 37 °C. Once the medium was aspirated, samples were added (100 μL) and cells were incubated for 24 h at 37 °C. A volume of 20 μL of LPS (100 ng/mL) was added to stimulated cells. Nitrite accumulation, and indicator of nitric oxide (NO) synthesis, was measured in the macrophage culture medium by the Griess reaction following a previously described method [24]. Briefly, a mixture containing 100 μL of supernatant and 100 μL of Griess reagent [1% (*w/v*) sulfanyl amide and 0.1% (*w/v*) N-1-(naphthyl) ethylenediamine-di-HCl in 2.5% (*v/v*) H_3_PO_4_] was incubated for 15 min and the absorbance measured at 550 nm in a Synergy HTX microplate reader (BioTek Instruments, Inc.). A sodium nitrite standard curve (3.125–100 μM) was used to measure the amount of NO. Three independent experiments were conducted and data were expressed as the mean and standard deviation (SD) (*n* = 12).

### 2.7. Statistical Analysis

All data were analyzed in three independent experiments, and results were expressed as the mean ± SD. Data were analyzed using one-way analysis of variance (ANOVA). All analyses were run with the program GraphPad Prism v.9.0.1 (GraphPad Software, San Diego, CA, USA).

## 3. Results and Discussion

### 3.1. Characterization of Chachafruto Seed Protein Concentrate (CPC): Behavior under Simulated Gastrointestinal Digestion

The moisture, lipid, ash, and fiber content of CPC was 9.78, 0.20, 3.64, and 1.42% (dry weight), respectively. The protein content of CPC was 82.5%, higher than the contents of 65.43 and 62.00% reported by Intiquilla et al. [5] and Arango et al. [25], respectively. This variation could be due to the different protein content present in the cultivars resulting from the growing conditions and soil types [26]. The extraction conditions could also contribute to the observed differences. Thus, alkaline extraction followed by acid precipitation used by Villafuerte et al. [27] resulted in higher protein content of the chachafruto isolate (88.99%), comparable to the value obtained in our study.

The composition of amino acids in CPC, expressed as g/100 g protein, is reported in Table 1. Seventeen amino acids were detected in the CPC, while tryptophan was not observed because of the analysis conditions that provoked the destruction of this amino acid by acid hydrolysis. Among the essential amino acids (EAA), lysine and leucine were the most abundant, with values of 4.00 ± 0.13 and 6.02 ± 0.17 g/100 g protein, respectively. Aspartic acid + asparagine and glutamic acid + glutamine were the most abundant among the non-essential amino acids (NEAA), with values of 8.49 ± 0.31 and 12.34 ± 0.36 g/100 g of protein, respectively. These values were higher than those reported by Intiquilla et al. [5] for the flour, concentrate and alcalase hydrolysate from *E. edulis*. Chachafruto amino acid profile has been reported to be similar to that reported for other legumes such as cowpea [28], peas and lentils [29], being deficient in sulfur amino acids, such as methionine and tryptophan. The EAA/TAA and EAA to NEAA ratios were 40% and 67%, respectively, similar to those proposed by FAO/WHO for the protein reference pattern (EAA/TAA: 40%; EAA/NEAA: 60%) [30]. CPC was subjected to gastrointestinal digestion simulating physiological conditions. The protein profile of CPC before and after the digestive process was evaluated by SDS-PAGE (Figure 1). This analysis revealed the presence of proteins with molecular weight ranged from 2 to 150 kDa in CPC (lane 2), with high intensity bands for 15–50 kDa proteins. The intensity of these bands decreased in GDCPC (lane 7) and GIDCPC (lane 11), indicating the partial susceptibility of chachafruto proteins to the action of digestive enzymes. Globulins and albumins are the major components of pulse proteins. Globulins are divided into two groups by their sedimentation coefficients, the 7 S fraction (vicilins) and 11 S fraction (legumins). The band ≈ 50 kDa might be assigned to vicilin, an oligomeric protein made of three polypeptide subunits (α, β, and γ) [31]. The polypeptides at 20 and ≤14 kDa visible in GDCPC and GIDCPC might correspond to fragments resulting from the action of digestive enzymes on vicilin [32].

The protein profiles of the fractions higher than 10 kDa obtained from GDCPC (lane 8) and GIDCPC (lane 12) were similar to those of their corresponding complete digests (lanes 7 and 11). In the case of 3–10 kDa fractions, only slight bands corresponding to medium-weight peptides released during the digestive process were visible (lanes 9, 13). No bands were visible for the fractions lower than 3 kDa (lanes 10, 14), possibly due to the gel and electrophoretic conditions used, which only allowed detecting high molecular weight peptides. However, the peptide mass distribution of the digests determined by MALDI-TOF-MS confirmed that gastric and pancreatic enzymes digested the chachafruto proteins in small peptide fragments with overall sizes under 3500 Da: 75.8% and 87.7% of peptides found in GDCPC and GIDCPC, respectively had a molecular weight 500–1500 Da, while peptides of 1500–3500 Da represented 34.5% in GDCPC and 18.9% in GIDCPC (Figure 2).

### 3.2. Impact of the Gastrointestinal Digestion on the Antioxidant Activity of Chachafruto Seed Protein Concentrate (CPC)

To evaluate the impact of the simulated digestive process on the release of bioactive peptides from chachafruto proteins, the antioxidant activity before and after digestion was studied by biochemical assays. At this time, the use of more than one method to study the antioxidant activity of a compound is suggested, evaluating its potentially mechanisms of action under different assay conditions [33]. In this study, the ABTS^•+^ and peroxyl (ROO^•^) radical scavenging activity of CPC, GDCPC, and GIDCPC and their ultrafiltered fractions were analyzed (Table 2).

The ABTS assay measured the suppressive capacity of an antioxidant peptide against the radical ABTS^•+^. When added to medium containing ABTS^•+^, the peptides may act as electron or hydrogen donors, transforming this radical cation into the non-radical ABTS [34]. Although CPC showed slight capacity to scavenge the radical (TEAC value of 0.10 µmol TE/mg protein), the values notably increased after the action of gastric and mainly, pancreatic enzymes, reaching a TEAC value of 0.46 µmol TE/mg protein in GIDCPC. This value is within the range (0.35–0.91 µmol TE/mg protein) described by Intiquilla et al. [5] for CPC hydrolyzates by Neutrase^®^ and Alcalase^®^. Same behavior was observed against the ROO^•^ radical, increasing the ORAC value up to 140% when chachafruto proteins were digested under gastrointestinal conditions. The ORAC assay is a broadly accepted method based on ROO^•^ scavenging for assessing food extracts that contain various antioxidants. It provides a single measurement of the inhibition time and degree. This assay uses the ROO^•^ radical as the reactant, with the redox potential and reaction mechanism similar to those of physiological oxidants. Another advantage of this method is the selection of physiological pH that allows the antioxidants reacting with an overall charge and protonation state similar to that occurring in the body [35]. The main contributors on the radical scavenging capacity observed in our study were low molecular weight peptides. Thus, the TEAC and ORAC values of the fraction F ˂ 3 kDa were 0.79 µmol TE/mg protein and 2.20 µmol TE/mg protein, respectively (Table 2). These results agree with previous studies carried out with animal protein-derived fractions that demonstrated that peptides lower than 3 kDa showed the greatest antioxidant activity [36]. However, contradictory studies found the highest antioxidant activity in fractions containing high molecular weight peptides [37]. The highest activity of low molecular weight peptides might be due to their improved accessibility to the redox reaction system or the degree of freedom of the critical amino acid residues [38]. In addition to peptide length, amino acid composition is another contributor to the antioxidant activity of food-derived peptides. Thus, the presence of hydrophobic amino acids has been associated with the antioxidant properties of peptides [39,40]. In the CPC, HAA represent 40.2% of TAA (Table 1), thus, they could contribute on the antioxidant activity of released peptides after gastrointestinal digestion.

### 3.3. Effect of Chachafruto Seed Protein Concentrate (CPC) on Macrophage RAW264.7 Cells under Basal and Stimulated Conditions

A macrophage cell model was used to evaluate the antioxidant and immunomodulatory effects of chachafruto protein-derived peptides under basal and stimulated conditions. Firstly, the effect of CPC and its gastric and gastrointestinal digests on cell viability was evaluated after 24 h treatment at concentrations of 5–200 µg protein/mL (Figure 3). LPS (100 ng/mL), used as positive control, significantly reduced the cell viability up to 86.62 ± 6.53%. A significant and dose-dependent reduction was also observed when basal cells were treated with CPC (at doses higher than 10 µg protein/mL), indicating the potential cytotoxic effects of chachafruto proteins (Figure 3A). After hydrolysis by digestive enzymes, the cytotoxic effects of CPC were partially reduced, and cell viability inhibition was observed when cells were treated with CPC digests at doses higher than 50 µg protein/mL). Moreover, an induction of the cell viability of macrophages was observed when cells were treated with 5 µg protein/mL. This protective effect at low doses was also observed in LPS-challenged macrophages (Figure 3B). This result was consistent with previous studies that reported protective effects of natural bioactive compounds or food extracts against apoptosis induction by LPS through the modulation of different molecular pathways [41].

LPS has multiple effects on macrophages, including the induction of a variety of inflammatory modulators such as interleukin (IL)-1, tumor necrosis factor (TNF-α), and NO, among others [42]. A large amount of NO, particularly synthesized by iNOS, stimulates an inflammatory response to inhibit the growth of invading microorganisms and tumor cells [43]. Figure 4 summarizes the results obtained by exposing macrophages to CPC, GDCPC, GIDCPC and their fractions higher than 10 kDa. Under basal conditions, macrophages did not produce NO. However, challenging with LPS at 100 ng/mL resulted in a significant increase of released NO (49.83 µM ± 3.96), as has been previously reported [44]. Dose-dependent NO induction was also observed when basal cells were treated with CPC, reaching 80.35 µM ± 4.57 at 100 µg protein/mL (*p* < 0.001) (Figure 4A). Among ultrafiltered fractions, those containing medium and large peptides were the major contributors to the NO induction caused by CPC. Thus, at 100 µg protein/mL, the NO levels measured in cells treated with ˃10 kDa and 3–10 kDa fractions were 24.31 µM ± 2.11 (Figure 4A) and 39.13 µM ± 0.97, respectively. However, treatment of cells with a fraction lower than 3 kDa did not result in any increase in NO levels (data not shown) unlike that reported by Li and coworkers [45]. These authors found that the <3 kDa peptide in *Tricholoma matsutake* water extract induced NO production by macrophages at doses ranging from 100 to 400 µg/mL. In basal cells treated with GDCPC and GIDCPC, the increase in NO production was also dose-dependent, although the maximum reached at 100 µg protein/mL was lower than that determined in CPC-treated cells (Figure 4C,E). This result suggests that undigested proteins and polypeptides were responsible for the NO-inducing properties of CPC. In the digests, only fraction ˃10 kDa induced NO release, confirming the ability of high molecular weight peptides to exert an immunostimulant effect on macrophage cells.

In challenged cells, CPC potentiated the effect of LPS at a dose-dependent manner, reaching NO levels of 80.75 µM ± 6.05 (Figure 4B), similar to that determined in non-induced and CPC-treated cells. However, its fraction ˃ 10 kDa at 10 µg protein/mL reduced the stimulating effect caused by LPS, without exerting any effect at higher doses. The reverting effect was also observed when macrophages were co-incubated with GDCPC and LPS, reducing NO levels up to 15.45 µM ± 1.35 when the gastric digest was used at 10 µg protein/mL (Figure 4D). No effects were observed on LPS-induced cells when GDCPC was used at doses higher than 50 µg protein/mL or when cells were co-incubated with LPS and its fraction ˃ 10 kDa. However, both GIDCPC and its high molecular weight peptide fraction potentiated the NO-inducing effect caused by LPS (Figure 4F). These results indicate that peptides contained in GDCPC could protect cells from the LPS challenge while high molecular weight peptides contained in CPC and its gastrointestinal digest activated macrophages to release inflammatory mediators such as NO. This is a free radical capable of reacting with superoxide anion to generate a selective oxidant and nitrating agent, peroxynitrite, that interacts with biological molecules and damages the cell membranes, being able to result in cell death [46]. Although properly regulated inflammatory responses are necessary for healthy immunofunction, an excessive inflammation may cause a chronic inflammatory status, sepsis, and even death [47]. Numerous peptides from legumes are considered to potentially work as antioxidant and anti-inflammatory agents through quenching free radicals, increasing antioxidant defenses, or inhibiting the release of proinflammatory mediators such as NO. As an example, digests from lupine protein contained different peptides in the extruded lupin seed flour fraction 2 (ELPF-2), which significantly reduced the production of NO and other mediators like TNF-α, IL-1β in macrophages [48]. Immunomodulatory capacities of food-derived peptides have been reported to be affected by their amino acid composition, length, sequence, and molecule structure [49].

To evaluate the effects of chachafruto protein-derived peptides on the oxidative status of cells, ROS levels were measured using the DCFH-DA probe in RAW264.7 cells under basal conditions (Table 3). The evaluation of intracellular ROS content is considered a good indicator of the oxidative damage in living cells [50]. LPS (100 ng/mL), as positive control, induced ROS generation (186.12% ± 16.43) as previously reported [51]. CPC, GDCPC and its fraction ˃10 kDa exerted a prooxidant effect at high doses. However, antioxidant activity was observed when cells were treated with these samples at low doses (Table 3). Fractions containing medium and low molecular weight peptides exerted protective effects at all assayed doses, indicating that these peptides were the main responsible for the alleviating effects on the oxidative status. Similar protective effects against oxidative stress were reported for the low molecular weight peptides extracted from the edible fungi *T. matsutake* [45]. Our results were in agreement with the observed in vitro antioxidant activity. However, in LPS-challenged cells, the decrease in ROS production occurred only in CPC F < 3 kDa at 100 µg/mL and GIDCPC F 3–10 kDa at 50 µg/mL and F < 3 kDa at 10 µg/mL (data not shown). Peptide fractions and peptides isolated from legumes have been reported to possess antioxidant properties in cell models. Martínez-Leo et al. [52], assessed the antioxidant and protective effect of the peptide fractions derived from *Mucuna pruriens*. In this study, the highest antioxidant activity was exhibited by the 1–3 kDa fraction that inhibited ROS production in H_2_0_2_-treated HeLa cells, lowering the intracellular levels by 207 ± 4.20%. Hernández-Ledesma et al. [53], demonstrated the capacity of soybean peptide lunasin to decrease ROS levels in LPS-activated macrophages at concentrations ranged from 50 µM to 200 µM. Also, the release of antioxidant peptides during simulated gastrointestinal digestion of red bean protein concentrate was reported by Piñuel et al. [54]. These authors demonstrated that after the action of gastric and pancreatic enzymes, peptides released to red bean proteins were able to reduce the intracellular ROS levels up to 75.30% and 66.40%, respectively.

## 4. Conclusions

After the action of gastric and pancreatic enzymes, the protective effects of chachafruto proteins against oxidative stress were induced through the capacity shown by released low molecular weight peptides to scavenge radicals. Moreover, high molecular weight peptides contained in the CPC and GIDCPC were able to activate macrophages to release immunoresponse-associated mediators such as NO, while peptides contained in the GDCPC could protect cells from LPS challenge, reducing that biomarker. In conclusion, chachafruto proteins can be considered an alternative protein source with promising nutritional and biological properties after degradation in the gastrointestinal tract, releasing essential amino acids and bioactive peptides. Further research focused on the identification of peptides responsible for the beneficial effects and the confirmation of their bioavailability should be conducted to promote the use of these proteins as a new antioxidant and immunostimulatory ingredient in functional foods and nutraceuticals.

## Figures and Tables

**Figure 1 nutrients-14-05256-f001:**
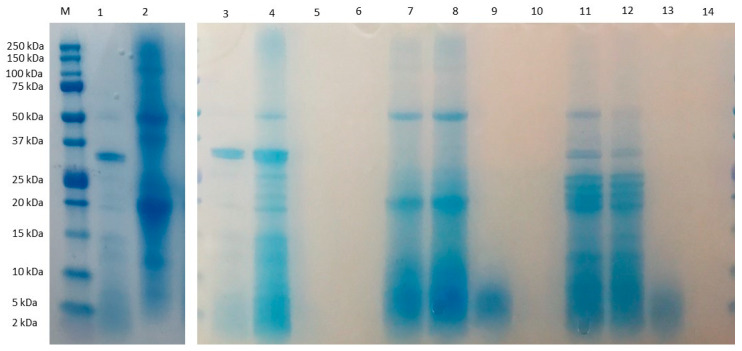
Electrophoretic (SDS-PAGE) analysis of *Erythrina edulis* (chachafruto) protein concentrate from seed (CPC) before and after its digestion simulating gastrointestinal conditions, and fractions collected from digested CPC by ultrafiltration. (M) molecular marker; (1,3) digestion blank (DB); (2) CPC; (4) fraction ˃ 10 kDa from DB; (5) fraction 3–10 kDa from DB; (6) fraction ˂ 3 kDa from DB; (7) gastric digest from CPC (GDCPC); (8) fraction ˃ 10 kDa from GDCPC; (9) fraction 3–10 from GDCPC; (10) fraction ˂ 3 kDa from GDCPC; (11) gastrointestinal digest from CPC (GIDCPC); (12) fraction ˃ 10 kDa from GIDCPC; (13) fraction 3–10 kDa from GIDCPC; (14) fraction ˂ 3 kDa from GIDCPC.

**Figure 2 nutrients-14-05256-f002:**
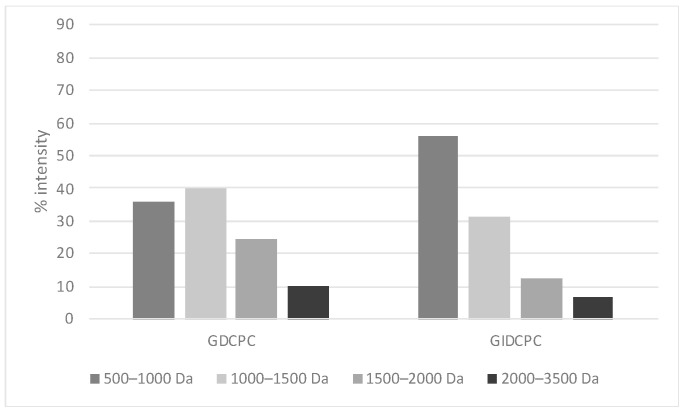
Peptide mass distribution (% intensity) of the gastric and gastrointestinal digests of chachafruto seeds protein concentrate (GDCPC and GIDCPC).

**Figure 3 nutrients-14-05256-f003:**
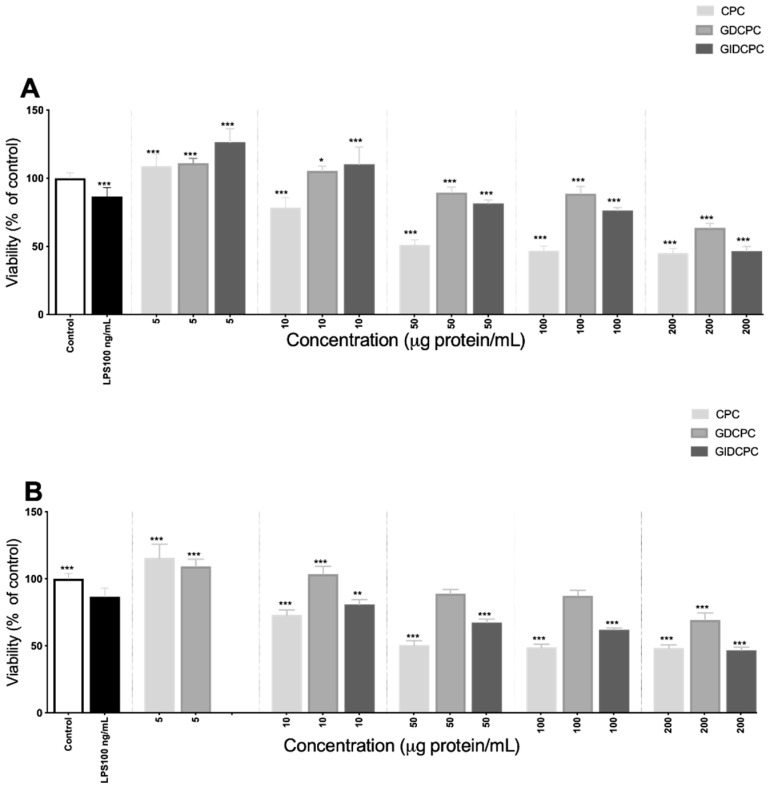
Viability (expressed as % of control considered as 100% cell viability) of (**A**) basal RAW264.7 macrophages and (**B**) lipopolysaccharide (LPS)-stimulated RAW264.7 macrophages, after 24 h incubation with different concentrations (5–200 µg protein/mL) of undigested and digested chachafruto seed protein concentrate (CPC, GDCPC, and GIDCPC). Significant differences compared to the control (* *p* < 0.05; ** *p* < 0.01; *** *p* < 0.0001).

**Figure 4 nutrients-14-05256-f004:**
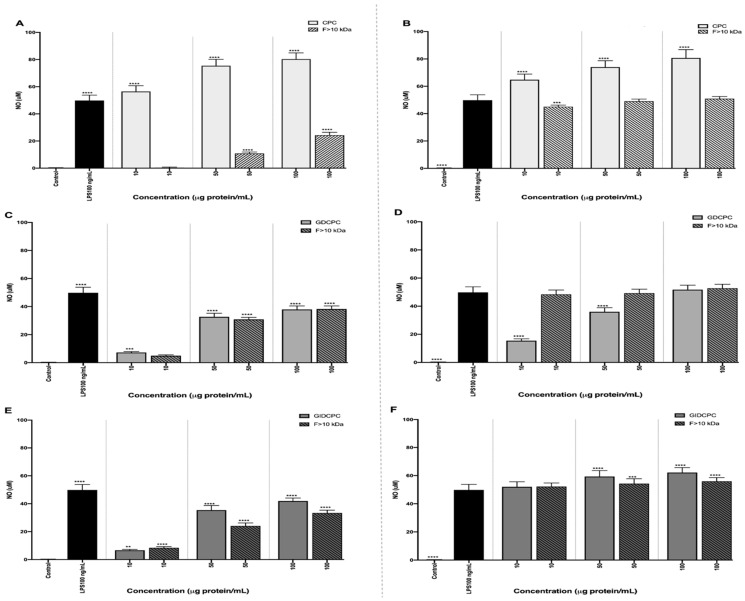
Nitric oxide (NO) production (μM) by (**A**,**C**,**E**) basal and (**B**,**D**,**F**) LPS-stimulated RAW264.7 macrophages after 24 h of exposure to (**A**,**B**) chachafruto protein concentrate (CPC) and its fraction higher than 10 kDa, (**C**,**D**) gastric digest from CPC (GDCPC) and its fraction higher than 10 kDa, (**E**,**F**) gastrointestinal digest from CPC (GIDCPC) and its fraction higher than 10 kDa. Significant differences compared to the control (** *p* < 0.01; *** *p* < 0.001; **** *p* < 0.0001).

**Table 1 nutrients-14-05256-t001:** Amino acid composition (g/100 g protein and g/100 g product) of *Erythrina edulis* protein concentrate from seed (CPC).

Amino Acid	Amino Acid Content (g/100 g Protein)	Amino Acid Content (g/100 g Product)	FAO Recommendations
Essential			
Lysine	4.00 ± 0.13	3.30 ± 0.11	5.20
Tryptophan	n.d.	n.d.	0.70
Phenylalanine	3.38 ± 0.10	2.78 ± 0.08	4.60 ^a^
Tyrosine	3.76 ± 0.08	3.10 ± 0.06	
Methionine	1.01 ± 0.08	0.83 ± 0.06	2.60 ^b^
Cysteine	0.90 ± 0.14	0.74 ± 0.12	
Threonine	2.56 ± 0.06	2.11 ± 0.05	2.70
Leucine	6.02 ± 0.17	4.97 ± 0.14	6.30
Isoleucine	2.32 ± 0.08	1.91 ± 0.06	3.10
Valine	3.23 ± 0.02	2.67 ± 0.01	4.20
Non-essential			
Aspartic acid + asparagine	8.49 ± 0.31	7.00 ± 0.26	
Glutamic acid + glutamine	12.34 ± 0.36	10.18 ± 0.30	
Serine	4.69 ± 0.11	3.87 ± 0.09	
Histidine	1.68 ± 0.10	1.38 ± 0.08	
Arginine	3.52 ± 0.18	2.90 ± 0.15	
Alanine	2.80 ± 0.13	2.31 ± 0.11	
Proline	3.83 ± 0.09	3.16 ± 0.08	
Glycine	3.28 ± 0.15	2.70 ± 0.12	
TAA	67.81	55.91	
HAA	27.25	22.48	
AAA	7.13	5.89	

n.d. not determined; ^a^ phenylalanine + tyrosine; ^b^ methionine + cysteine; HAA: hydrophobic amino acids (alanine, valine, isoleucine, leucine, tyrosine, phenylalanine, tryptophan, methionine, proline, and cysteine); TAA: total amino acids; AAA: aromatic amino acids (phenylalanine, tryptophan, and tyrosine). Data are the mean of two determinations.

**Table 2 nutrients-14-05256-t002:** Antioxidant activity (ABTS and ORAC) of *Erythrina edulis* protein concentrate from seed (CPC), its gastric (GDCPC) and gastrointestinal (GIDCPC) digests and their fractions obtained by ultrafiltration.

Sample	TEAC (µmol TE/mg Protein)				ORAC (µmol TE/mg Protein)			
	Whole Sample	F ˃ 10 kDa	F 3–10 kDa	F ˂ 3 kDa	Whole Sample	F ˃ 10 kDa	F 3–10 kDa	F ˂ 3 kDa
CPC	0.10 ± 0.01	0.20 ± 0.00	0.47 ± 0.01	0.69 ± 0.07	0.47 ± 0.02	0.70 ± 0.01	0.79 ±0.01	4.29 ± 0.13
GDCPC	0.33 ± 0.02	0.17 ± 0.01	0.34 ± 0.00	0.66 ± 0.06	1.13 ± 0.02	0.47 ± 0.02	1.10 ± 0.02	1.93 ± 0.13
GIDCPC	0.46 ± 0.01	0.21 ± 0.01	0.30 ± 0.02	0.79 ± 0.00	1.12 ± 0.07	0.68 ± 0.02	0.96 ± 0.02	2.20 ± 0.18

TEAC: trolox equivalent antioxidant capacity; TE: trolox equivalent; ORAC: oxygen radical antioxidant capacity.

**Table 3 nutrients-14-05256-t003:** Effects of *Erythrina edulis* protein concentrate from seed (CPC), its gastric (GDCPC) and gastrointestinal (GIDCPC) digests, and their fractions obtained by ultrafiltration on reactive oxygen species (ROS) generation in macrophage RAW264.7 cells under basal conditions.

Sample	Concentration (µg/mL)	ROS Generation (% of Control)			
		Whole Sample	Fraction ˃ 10 kDa	Fraction 3–10 kDa	Fraction ˂ 3 kDa
Control	-	100.00 ± 4.87			
LPS	0.1	186.12 ± 16.43 ***			
CPC	5.0	65.55 ± 3.69 ***	85.73 ± 6.26 *	57.03 ± 4.40 ***	52.84 ± 4.43 ***
	10.0	102.56 ± 10.56	116.25 ± 6.47 *	58.14 ± 4.44 ***	50.54 ± 3.28 ***
	50.0	140.62 ± 8.44 ***	188.80 ± 8.90 ***	60.21 ± 3.85 ***	57.26 ± 5.61 ***
	100.0	163.20 ± 15.91 ***	225.02 ± 6.63 ***	66.48 ± 5.97 ***	61.19 ± 3.34 ***
GDCPC	5.0	41.96 ± 5.04 ***	70.56 ± 6.00 ***	61.70 ± 2.30 ***	58.80 ± 2.69 ***
	10.0	50.76 ± 6.12 ***	75.00 ± 5.17 ***	57.97 ± 2.28 ***	60.67 ± 2.97 ***
	50.0	115.53 ± 10.16 *	199.00 ± 19.47 ***	82.02 ± 5.62 ***	70.23 ± 3.21 ***
	100.0	137.26 ± 8.69 ***	236.38 ± 22.35 ***	118.59 ± 5.92 **	75.19 ± 4.69 ***
GIDCPC	5.0	67.56 ± 3.77 ***	66.30 ± 4.86 ***	53.76 ± 4.38 ***	61.02 ± 2.51 ***
	10.0	74.00 ± 6.83 **	106.74 ± 10.26	55.42 ± 5.23 ***	66.99 ± 2.13 ***
	50.0	224.44 ± 19.08 ***	180.46 ± 14.09 ***	68.96 ± 3.13 ***	73.91 ± 4.23 ***
	100.0	306.17 ± 26.57 ***	245.40 ± 8.98 ***	70.23 ± 5.56 ***	77.81 ± 2.30 ***

Data indicate mean value ± standard deviation. Significant differences compared to the control (* *p* < 0.01; ** *p* < 0.001; *** *p* < 0.0001).

## Data Availability

Not applicable.

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
