# Peer review of "Release of Bioactive Peptides from Erythrina edulis (Chachafruto) Proteins under Simulated Gastrointestinal Digestion"

_nutrients, 2022, doi:10.3390/nu14245256_

Round 1
Reviewer 1 Report
The manuscript evaluated the release of bioactive peptides from the proteins of Erythrina edulis (Chachafruto) under simulated gastrointestinal digestion. The alternative protein was further evaluated for its potential antioxidant activities using known and established assays.
General comments
The manuscript is well-written and presents an introduction with a good background. The methods are adequately described, and the results are presented and discussed clearly. Also, the conclusions are supported by the results, and the research novelty is well presented throughout the manuscript. Some aspects are detailed below to help authors to improve the manuscript for possible publication:
Abstract: The abstract is well presented. However, the author needs to include some of the results in the abstract. For example, the results of the antioxidant activities and other important findings should be included in the abstract.
Introduction: The Introduction section is well-written and theoretically well-founded. The research novelty is well-emphasized. I suggest some minor corrections in this section, as follows:
Line 39: Include at least two references here
Line 41: Include citation after medicinal uses
Line 47: change this to “to the best of our knowledge, there are no data about”
Materials and methods:
The materials and methods section is detailed enough.
Results and discussion:
Line 224: Write the amino acid name in full throughout the manuscript.
Line 237: How did you know the molecular weight range of proteins in the samples?
The result and discussion section are too short. A lot of important information is missing, and the authors need to elaborate on every important point mentioned in the manuscript. In addition, references should be provided where necessary.
Author Response
The manuscript evaluated the release of bioactive peptides from the proteins of Erythrina edulis (Chachafruto) under simulated gastrointestinal digestion. The alternative protein was further evaluated for its potential antioxidant activities using known and established assays.
General comments
The manuscript is well-written and presents an introduction with a good background. The methods are adequately described, and the results are presented and discussed clearly. Also, the conclusions are supported by the results, and the research novelty is well presented throughout the manuscript. Some aspects are detailed below to help authors to improve the manuscript for possible publication:
Abstract: The abstract is well presented. However, the author needs to include some of the results in the abstract. For example, the results of the antioxidant activities and other important findings should be included in the abstract.
Answer: As suggested by the reviewer, the abstract has been modified to include the activities that have been demonstrated for the pajuro protein digests.
Introduction: The Introduction section is well-written and theoretically well-founded. The research novelty is well-emphasized. I suggest some minor corrections in this section, as follows:
Line 39: Include at least two references here
Answer: The recent reference “Otero, D.M.; Mendes, G.D.L.; Lucas, A.J.D.; Christ-Ribeiro, A.; Ribeiro, C.D.F. Exploring alternative protein sources: Evidence from patents and articles focusing on food markets. Food Chem. 2022, 394, 133486” has been added (line 41, reference 3).
Line 41: Include citation after medicinal uses
Answer: The reference “Sánchez Chero, M.J.; Sánchez Chero, J.A.; Miranda Zamora, W. Technify and conserve the bioactive components of Pashul (Erythrina edulis) for human consumption. UCV HACER Rev. Inv. Cult. 2019, 8, 11–17” has been added (line 43, reference 4).
Line 47: change this to “to the best of our knowledge, there are no data about”
Answer: The sentence has been changed as suggested (line 49).
Materials and methods:
The materials and methods section is detailed enough.
Results and discussion:
Line 224: Write the amino acid name in full throughout the manuscript.
Answer: The full name of amino acids has been written along the manuscript as indicated by the reviewer.
Line 237: How did you know the molecular weight range of proteins in the samples?
Answer: In the SDS-PAGE analysis, a molecular weight protein marker is used (lane M in the gel). Since the molecular weight of the proteins contained in the marker are known, it is possible to extrapolate the approximately molecular weight of the proteins contained in the samples.
The result and discussion section are too short. A lot of important information is missing, and the authors need to elaborate on every important point mentioned in the manuscript. In addition, references should be provided where necessary.
Answer: Some sentences have been added to the results and discussion section as suggested by the reviewer (marked in red color). Also, several references (36, 44, 45, and 49) have been included.
Reviewer 2 Report
The manuscript of Correa et al. reports the potential of the Erythrina edulis (chachafruto) protein. In particular, the authors demonstrate that these proteins release multifunctional peptides after their transit through the gastrointestinal tract. These released low molecular weight peptides showed protective effects against oxidative stress mediated by their radical scavenging activity, while high molecular weight peptides exerted immunostimulating effects through upper-regulation of immune response-associated biomarkers.
In mi opinion the manuscript is interesting but there are some points to be clarified for its publication.
Main point
1. In paragraph 2.3 lines 99-100 the authors write “change chromatography using a Biochrom 30 series Amino Acid Analyser (Biochrom, Cambridge, US) after automatic pre-column derivatization of samples with ninhydrin and”. This is a serious conceptual error. The determination of amino acids with ninhydrin requires a post column reaction!. Without reading previous works (cup and paste), they can read the work of the Nobel prize winners Stein and Moore (1954).
2. The SDS-PAGE electropherogram reported in figure 1 is not a clear image, but it looks like a puzzle of different pictures. I ask for the raw images for a check. Moreover, lane 4 and 6 are very similar? how come.
3. In the preparation of the hydrolysates there is no mention of the inactivation of the proteolytic enzymes used. Although pepsin inactivates at neutral pH, this is not the case with the other enzymes used.
Minor point:
1. Page 2, line 60: “imm3unomodulatory” should be corrected perhaps it is a typo due to “a cut and paste action”
2. Page 2, please delete “Obtention” from the title of paragraph 2.2 and 2.3.
Finally, even if I'm not an expert, English should be reviewed by an expert.
Author Response
The manuscript of Correa et al. reports the potential of the Erythrina edulis (chachafruto) protein. In particular, the authors demonstrate that these proteins release multifunctional peptides after their transit through the gastrointestinal tract. These released low molecular weight peptides showed protective effects against oxidative stress mediated by their radical scavenging activity, while high molecular weight peptides exerted immunostimulating effects through upper-regulation of immune response-associated biomarkers.
In mi opinion the manuscript is interesting but there are some points to be clarified for its publication.
Main point
1. In paragraph 2.3 lines 99-100 the authors write “change chromatography using a Biochrom 30 series Amino Acid Analyser (Biochrom, Cambridge, US) after automatic pre-column derivatization of samples with ninhydrin and”. This is a serious conceptual error. The determination of amino acids with ninhydrin requires a post column reaction!. Without reading previous works (cup and paste), they can read the work of the Nobel prize winners Stein and Moore (1954).
Answer: We want to thank the reviewer by the comment. It has been a mistake. The derivatization with ninhydrin is post-column. This has been corrected (line 100).
2. The SDS-PAGE electropherogram reported in figure 1 is not a clear image, but it looks like a puzzle of different pictures. I ask for the raw images for a check. Moreover, lane 4 and 6 are very similar? how come.
Answer: The samples were run in two different gels at the same time due to the lack of space to include all samples. That is the reason why the gels look different. We have substituted the figure by that directly obtained after staining with Coomassie Blue without changing the background color. The legend has been modified accordingly. As indicated into the manuscript, the profiles of fractions higher than 10 kDa were similar to those of whole digests, only observing variations in the intensity of the bands.
3. In the preparation of the hydrolysates there is no mention of the inactivation of the proteolytic enzymes used. Although pepsin inactivates at neutral pH, this is not the case with the other enzymes used.
Answer: The gastric enzyme pepsin is inactivated by raising the pH. In the case of pancreatic enzymes, their inactivation is achieved by heating (80˚C for 15 minutes). This information was included into the manuscript (lines 125-126).
Minor point:
1. Page 2, line 60: “imm3unomodulatory” should be corrected perhaps it is a typo due to “a cut and paste action”
Answer: In our computer, the word looks well written.
2. Page 2, please delete “Obtention” from the title of paragraph 2.2 and 2.3.
Answer: As suggested by the reviewer, the titles of sections 2.2. and 2.3 have been modified.
Finally, even if I'm not an expert, English should be reviewed by an expert.
Answer: The English language has been revised.